# King's Sarcoidosis Questionnaire (KSQ) – Validation study in Serbian speaking population of sarcoidosis patients

Mihailo Stjepanovic[1,2]*, Violeta Mihailovic-Vucinic[2], Branislav S. Gvozdenovic[3], Jelena Milin-Lazovic[4], Slobodan Belic[1], Natasa Djurdjevic[1], Nikola Maric[1], Aleksa Golubovic[1]

1 Clinic of Pulmonology, University Clinical Center of Serbia, Belgrade, Serbia, 2 Faculty of Medicine, University of Belgrade, Belgrade, Serbia, 3 Pharmacovigilance Department, PPD Serbia, Belgrade, Serbia, 4 Institute for Medical Statistics and Informatics, Faculty of Medicine, University of Belgrade, Belgrade, Serbia

* mihailostjepanovic@gmail.com

## Abstract

### Introduction

Sarcoidosis is a multiorgan, multisystem chronic disease of unknown etiology and unpredictable course. Health status is reduced in sarcoidosis and assessing it is a difficult multi-task effort due to many faces this disease might have. Recently, a new questionnaire for assessing health status in sarcoidosis was developed by a group of authors from England–King's Sarcoidosis Questionnaire (KSQ). The benefit of KSQ is the ability to develop the best care plan for the patient, as well as to differentiate the efficacy of the administered treatment.

### Objective

The aim of this study was to validate the KSQ in Serbian speaking population of sarcoidosis patients. The test itself is a modular, multi-organ health status measure for patients with sarcoidosis for use in clinic and the evaluation of therapies. The correlation of KSQ with different clinical course of sarcoidosis (acute vs chronic disease) and with the clinical outcome status (COS) in sarcoidosis was also investigated.

### Methods

A total of 159 biopsy positive sarcoidosis patients participated in this study. The average age of the participants was 49.67, majority was female (67.3%) and majority had only pulmonary form of sarcoidosis (71.7%). KSQ - new disease-specific health status instrument, was compared with 5 other already existing instruments already used and validated in sarcoidosis (Saint George Respiratory Questionnaire- SGRQ, Daily Activity List -DAL, Fatigue Assessment Scale- FAS, Medical Research Council dyspnea scale–MRC, Borg Dyspnea Scale and 15D as general questionnaire.

**Data Availability Statement:** All relevant data are within the paper and its Supporting Information files.

**Funding:** The author(s) received no specific funding for this work.

**Competing interests:** The authors have declared that no competing interests exist.

## Results

KSQ has significant correlation with other quality of life questionnaires already used in sarcoidosis. Translated version of KSQ shows significant internal reliability, similar to the original KSQ. Serbian version of KSQ has significant correlation with different clinical course of sarcoidosis and with COS as well. The translated version of KSQ is reliable sarcoidosis specific instrument for assessing health status in these patients.

## Introduction

Sarcoidosis is a multisystemic inflammatory disease of unknown origin, in which non-caseous granuloma present in different organs leads to decrease of function. Depending on the localization and extensivity of granuloma, sarcoidosis can present with a wide variety of signs and symptoms [1].

The most commonly afflicted organ in sarcoidosis are lungs and mediastinal lymph nodes, therefore, performing pulmonary function tests (PFTs) and chest X ray are essential in both diagnostics and follow-ups. Being easily performed and replicated, both procedures can help the physician evaluate the activity and severity of the disease [2]. It should be noted that the clinical presentation of sarcoidosis is not specific, and chief complaints in patients are fatigue, chest pain, cough, dyspnea, fever and swelling of joints. The presentation is further complicated when taking into consideration the chronic and extrapulmonary forms of disease. The available tests and the severity of symptoms have not shown correlation, which has caused a certain division between physicians and patients regarding the intensity of presented symptoms [3]. One of the most commonly presented symptoms in patients with sarcoidosis is fatigue, which is completely subjective parameter, and, if left untreated, can lead to significant cognitive impairment [4, 5].

As sarcoidosis most commonly afflicts the working age population, any prolonged activity of the disease and its symptoms leads to significant decrease of quality of life, which leads to further deterioration of the state of the individual. Previous studies have tried to find a correlation between the intensity of symptoms and the prognosis of the disease [6, 7], however, no precise algorithm could be developed. It should be noted that patients with no symptoms at the moment of diagnostics have better clinical outcome, compared to patients with any clinical presentation.

The relationship between heath status and sarcoidosis was first examined in 1997 [8]. Since that time until now authors have been trying to find out the best measure considering patients quality of life and/or health status [9].

The need for a quality-of-life questionnaire specific for sarcoidosis has led to the development of King's Sarcoidosis Questionnaire (KSQ) [10]. KSQ is a modular multi-organ health status measure for patients with sarcoidosis for use in clinic and the evaluation of therapies. It consists of five modules: General health status-GHS (10 items), Lung (6 items), Medication (3 items), Skin (3 items) and Eye (7 items). The General health status module is intended to be administered to all patients with sarcoidosis. In addition to this, patients also complete organ specific modules if relevant to their condition. The individual module scores are intended to identify the health domains affected. The medication module can be used in isolation or combined with overall lung and skin health status questionnaires but not eye health status. The completion of the test consists of patients filling out the original seven-point Likert scale and scoring is calculated using a re-ordered scale for appropriate items. The KSQ module and

overall (total) scores were transformed to a range of 0–100 [(actual score − lowest possible score/range) ×100], where 100 represents the best health status [10].

The aim of this study was to validate the KSQ in Serbian speaking population of sarcoidosis patients. The objective was also to compare the KSQ as disease-specific health status questionnaire with other already used questionnaires in sarcoidosis.

## Material and methods

The study was conducted in the Clinic of Pulmonology, University Clinical Center of Serbia in Belgrade, Serbia, over the period of 6 months. Prior to the start of the study, the study had been presented to the Ethic committee of the Clinic of Pulmonology, University Clinical Center of Serbia, and the permission had been given.

### Patients

A total of 159 biopsy positive sarcoidosis patients were enrolled into this study. All of the participants were diagnosed at the Clinic of Pulmonology of the University Clinical Center of Serbia, and were diagnosed by lung biopsy by bronchoscopy. The participation in the study was on voluntary basis, and prior to collecting the data, a written consent was acquired from the patients. All of the participants were of legal age (in Serbia 18-year-old and older), as our Clinic only treats adult patients, and did not have comorbidities that could impact the quality of life. The acquisition of all needed data was performed on their regular check-up, and the patients were chosen at random. Patients who had difficulty understanding the questions (such as illiterate patients), were not taken into consideration. Demographic data was collected, with classification of current organ involvement based on the ACCESS study [11]. Besides the demographic data, we also collected pulmonary function tests, other, previously validated tests, and the outcome of sarcoidosis.

### Pulmonary function tests

On the same day patients completed the KSQ, performed spirometry tests on a pneumotachograph (Jaeger, Germany) with measures expressed as % of the reference values according to the ATS/ERS criteria [12]. Pulmonary function measurements included forced expiratory vital capacity (FVC), forced expiratory volume in 1 second ($FEV_1$), peak expiratory flow (PEF). The transfer factor of the lung for carbon monoxide (DLCO) was measured using the single-breath method (Masterlab, Jaeger, Wurzburg, Germany).

### Questionnaires

During the regularly scheduled outpatient clinic visit patients completed the Serbian version of King's Sarcoidosis Questionnaire (KSQ), two standardized questionnaires for the measuring of health status: a generic measure–the fifteen-dimensional measure of health-related quality of life (15D) [13] and the Saint George Respiratory Questionnaire (SGRQ) [14], as well as Modified Medical Research Council (MRC) Dyspnea Scale, List of Daily Activities (DAL), and Fatigue Assessment Scale (FAS).

15D is an instrument for measurement of health-related quality of life [13]. It consists of 15 different and mutually exclusive health dimensions, each represented by one item [15, 16]. The total questionnaire score ranges between 0 and 1, where 1 signifies the highest level of health status. 15D was previously used in other diseases in multiple languages, and the Serbian version had been previously used in patients with asthma where it demonstrated good psychometric measurement properties [17].

SGRQ is an instrument that was originally designed to measure the health status of COPD patients [14]. Its validity, reliability, and responsiveness had been shown in other pulmonary diseases, including sarcoidosis [17]. The questionnaire consists of 50 items with 76 responses, and encompasses three domains of health status: 1) symptoms, with a special focus on respiratory symptoms; 2) activities, measuring decreased mobility or physical activity and 3) impacts, measuring the psychosocial influence of disease on the everyday life. Scores of these domains, as well as the total score, are scaled from 0 to 100, where higher scores represent higher decrease of quality of life.

Dyspnea was measured by the Modified Medical Research Council (MRC) Dyspnea Scale [18]. The MRC scale classifies subjects into one of five categories according to their degree of dyspnea while performing certain activities. Scores range from 0 to 4, with the higher scores indicating more severe dyspnea.

The degree of limitation in activities of daily living was evaluated with the List of Daily Activities (DAL) [19]. DAL has 11 items, all of which are related to everyday activities that a healthy individual can perform with no impairment. Higher the number of positive responses shows the higher degree of everyday impairment. DAL has been previously utilized in other chronic respiratory diseases, and the previously validated Serbian version has been used in patients with sarcoidosis [19, 20].

Fatigue Assessment Scale (FAS) is a scale specifically devised for measuring the fatigue, and can differentiate between physical and mental fatigue (5 items are related to physical, and 5 items are related to metal fatigue). The response scale is a 5-point scale (1 - never to 5 - always), and the score ranges between 10–50; the score bellow 22 is indicative of no fatigue.

As previously mentioned, all questionnaires except FAS have already been used in assessing symptom severity in Serbian population of sarcoidosis patients [21]. However, FAS has been validated in a broad spectrum of fatigue studies in sarcoidosis [4, 5]. The usage of other questionaries is to compare the results generated by previous, already validated, questionaries with KSQ, and, if any correlation was found, if one questionnaire (KSQ) can replace multiple questionnaires in everyday practice.

## Clinical outcome of sarcoidosis

The clinical outcome of sarcoidosis has been defined by WASOG (World Association of Sarcoidosis and other Granulomatous diseases) task force in several clinical phenotypes based on clinical outcome in sarcoidosis patients (COS). To determine COS category, the patients had to be checked up on for a period of 5 years, as of the diagnosing the sarcoidosis [23].

The participants have been separated in nine COS categories, further placed in five groups, based on the current or previous need for systemic therapy, the resolution of the disease, and current status of the condition. The estimated COS groups are shown in Table 1.

## Questionnaire translation process

The original English version of the KSQ was officially translated and adopted into the Serbian language by two bilingual experts, working independently in translation form English to Serbian and vice versa. The items have been standardized with the help of physicians from the Serbian Association of Sarcoidosis (SAS), and who had previous experience with the English version of KSQ.

The backward version of the KSQ translation was emailed to the authors of the original KSQ (S Birring and A Patel), alongside with the original version, and the KSQ scores were calculated by one of the original coauthors. After comparing the translated version with the original, the original authors have given the permission for use.

**Table 1. Clinical outcome in sarcoidosis patients.**

| | Never treated | No treatment for over a year | Asymptomatic | Symptomatic | Worsening in the last year |
|---|---|---|---|---|---|
| Resolved | COS 1 | COS 2 | | | |
| Minimal disease | COS 3 | COS 4 | | | |
| Persistent disease with no current treatment | COS 5 | COS 6 | | | |
| Persistent disease with current treatment and no worsening | | | COS 7 | COS 8 | |
| Persistent disease with current treatment | | | | | COS 9 |

COS- clinical outcome in sarcoidosis patients

After creating the final Serbian version, cognitive debriefing was performed; five patients and five physicians were chosen at random and were asked if they completely understand the questions from the KSQ. In addition, we asked both doctors and patients to write down their impressions about the new questionnaire. Furthermore, the doctors were asked to give a mark for the questionnaire at the scale from 0–5.

Here are doctors'comments about the KSQ, together with the marks.

## Doctors' comments about KSQ:

**Milica Kontic, MD, PhD**

I think that KSQ is clear, specific and very well formulated and that will be understandable to sarcoidosis patients and easy to fill out.

(5)

**Ana Blanka, MD**

I think that questionnaire is very good, because covers all aspects of sarcoidosis. Questions are clear, short and precise which makes this questionnaire understandable and easy to answer. Overview (appearance) is also very clear.

(5)

**Jasmina Maric Zivkovic, MD, PhD**

Symptoms from all sections are comprehensively (overall) presented. The questionnaire its self is very clear. The only remark: I would change is the order of questions in lung, medication and skin section. In medication and lung section the last question should be in the first place. I think that this sarcoidosis questionnaire is very good and useful. It's not very long and our patients shouldn't have any difficulties answering proposed questions.

(5)

**Snezana Raljevic, MD**

Questionnaire consists of 29 questions, divided into 5 subgroups: General health status, lungs, medications, skin and eyes. Questionnaire is short and clear. It wouldn't take too long for a patient to answer. Questions are understandable to patients and almost all conditions were included that could affect quality of life of sarcoidosis patient. Questions and answers are very well designed so the patients themselves could fill out this questionnaire without any help from investigators.

My opinion is positive for KSQ and I would recommend it for use.

(5)

**Aleksandra Dudvarski-Ilic, MD, PhD**

Questions absolutely reflect the patients' opinion about their disease. Almost all aspects of sarcoidosis are represented in this questionnaire that are important for patients. KSQ ic clear and overview is very good. Some terms in Serbian language could be more adjusted.

(4)

**Patients about KSQ**

Here we also considered patients'medical history, sarcoidosis duration and organ involvement. Additionally, patients were asked to give a mark for the questionnaire at the scale from 0–5.

**O.M/ Age 31**, sarcoidosis duration: 1-year, Current therapy: Prednisone

I think that this questionnaire includes most of the problems that sarcoidosis patients encounter. It's very clear and easy to answer.

(5)

**I.Dj/ Age 50**, sarcoidosis duration: 1-year, Current therapy: No

This questionnaire covers all symptoms of our disease, and I didn't have any difficulties answering these questions. I am also pleased that there are people who are trying to make our disease easier in this way.

(5)

**S.D/Age74**, sarcoidosis duration: 8 years, Current therapy: Prednisone+ Methotrexate

I have read this questionnaire very carefully and I think that includes a lot of symptoms that are related to sarcoidosis.

(5)

**S. P/Age 56,** sarcoidosis duration: 4 years, Current therapy: No

I think that based on my course of the disease this questionnaire covered the essence of sarcoidosis and all the symptoms and conditions sarcoidosis patient can experience.

(5)

**D.J/Age 42**, sarcoidosis duration: 5 years, Current therapy: Prednisone+ Methotrexate

Questions are precise, but I had hard time answering them. I believe the main lack of this questionnaire is that questions only consider 2-week period. I believe so, because while I am in a hospital, I rest all day and my symptoms are less there comparing when I am at home and having all my daily routine activities. I think real answers could be given when I am in my everyday environment.

(3)

It should be noted that the last patient was administered for the reevaluation of the disease, therefore the period which was not representative.

## Statistical analysis

Descriptive statistics were calculated for the baseline demographic and clinical features. Continuous variables were presented as means with standard deviations and 95% confidence intervals (CI), while categorical variables are presented with numbers and percentages. Normality of distribution for continuous variables was tested with mathematical and graphical methods, depending on the variable.

Construct validity was examine using explanatory factor analysis. Factor extraction was performed using the Principal Components Analysis method with Varimax rotation. Correlation between the general and organ specific domains of KSQ and the corresponding questionnaires were determined using Spearman's correlation coefficients. The internal consistency of the Serbian version of KSQ was assessed for multiple item scales by using Cronbach's alpha coefficient (ranges from 0–1, the latter meaning perfect reliability).

Differences between KSQ scores in groups with acute and chronic sarcoidosis were analyzed using Students t-test (or Mann Whitney test) as appropriate. Correlation between KSQ scores with COS score, other questionnaires scores and lung functions were calculated using Pearson (or Spearman) correlation coefficient as appropriate. Sample size for correlation was

calculated to detect at least correlation coefficient of 0.3, level of significance 0.05 and power of 80%. Therefore, the minimum required sample size for this study is 85. Recommended sample size for factor analysis is 5 to 10 cases per number of items. This rule taps the model precision, i.e., the ability of the parameter estimates to approximate true population values [22].

The level of significance was set at 0.05. Statistical analysis was performed using the IBM SPSS 21 (Chicago, IL, 2012) package.

## Results

Results for explanatory factor analysis were presented in Table 2. Kaiser-Meyer-Olkin test for sampling adequacy: 0.894 and highly significant (p<0.001) Bartlett's Test of Sphericity indicate that the application of factor analysis is adequate. By applying factor analysis, 5 factors were extracted, and which together explain 67.8% of the total variance. All five factors correspond to the five original modules. In the rotated factor matrix, item 1–7 and 10 items have the highest correlation with first domain, 12–16 with third, 20–22 with fifth, 23–25 with fourth, and 23–29 with second domain. Questions 8, 9 and 11 had highest correlation with fifth domain. The obtained structure corresponds to the original structure of the questionnaire with 5 domains and associated items. Correlation between the general and organ specific domains of KSQ and the corresponding questionnaires are presented in Table 3. Construct validity between the general and organ specific domains of KSQ and the corresponding questionnaires are presented in Table 3. Correlation between KSQ modules and SGRQ domains, DAL MRC, FAS and 15D scores was shown to be statistically significant, with the exception for medication and skin score. The correlations between the KSQ GHS and lung domain and all questionnaires (SGRQ domains, DAL MRC, FAS and 15D) were negative (r = -0.523 to -0.751). KSQ medication and skin score showed a negative, weak to moderate correlation with SGRQ domains, DAL MRC, FAS and 15D (r = -0.175 to - 0.350). The Eyes module showed a negative, moderate to strong correlation with the other questionnaires (r = -0.382 to -0.574). KSQ organ modules combined with the GHS module all showed a moderate to strong correlation with the SGRQ domains, DAL MRC, FAS and 15D.

Internal reliability for KSQ translated version and original version are showed in Table 4. Cronbach's α coefficients for lungs, skin and eye were higher in translated version of KSQ. Cronbach's α for general health status and medication were lower in Serbian version of KSQ, when compare to original version. The same coefficient was used in the validation of the original test. Overall Cronbach's α coefficient was 0.940.

Average age for patients with sarcoidosis was 49.67±11.12 and 2/3 were female. The baseline characteristics of the 159 patients who participated in the study are shown in Table 5.

Patients with chronic sarcoidosis showed statistically significant lower scores for all KSQ modules except for GHS and medication score. Also, GHS lung score was lower in chronic patients, difference was close to conventional level of significance (Table 6).

For patients with follow up period more than 5 years, clinical outcome status (COS) was analyzed as they were classified into 9 categories (Table 7).

Correlation between lung function parameters, clinical outcome of sarcoidosis and KSQ scores correlation are presented in Table 8. There was no statistically significant correlation between FVC and KSQ modules. FEV 1 correlate positive statistically significant only with lung score. Positive, statistically significant correlation was observed between PEF and DLco with GHS, Lung, Eye, GHS Lung, GHS Skin score, GHS Eye, GHS Lung Skin, GHS Lung Medication, GHS Skin Medication and GHS Lung Skin Medication score. Statistically significant negative correlation between COS score and KSQ lung, GSH Lung, GSH S, GSH LM, GSH SM, GSH LMS were observed. Negative, moderate statistically significant correlation was

**Table 2. Explanatory factor analysis-rotated component matrix.**

| question | Component | | | | |
|---|---|---|---|---|---|
| | 1 | 2 | 3 | 4 | 5 |
| 1 | **0.711** | 0.339 | 0.311 | 0.061 | 0.126 |
| 2 | **0.773** | 0.275 | 0.168 | 0.058 | 0.116 |
| 3 | **0.723** | 0.162 | 0.146 | 0.054 | 0.255 |
| 4 | **0.704** | 0.227 | 0.385 | 0.07 | 0.03 |
| 5 | **0.814** | 0.218 | 0.233 | 0.02 | 0.043 |
| 6 | **0.596** | 0.219 | 0.362 | 0.194 | -0.022 |
| 7 | **0.722** | 0.18 | 0.175 | 0.139 | 0.102 |
| 8 | 0.078 | 0.025 | 0.179 | -0.027 | **0.663** |
| 9 | 0.466 | -0.034 | 0.183 | 0.118 | **0.501** |
| 10 | **0.66** | -0.008 | 0.329 | 0.025 | 0.159 |
| 11 | 0.341 | 0.256 | 0.275 | 0.092 | **0.386** |
| 12 | 0.436 | 0.177 | **0.745** | 0.093 | 0.131 |
| 13 | 0.42 | 0.173 | **0.802** | 0.117 | 0.16 |
| 14 | 0.49 | 0.214 | **0.678** | 0.035 | 0.188 |
| 15 | 0.39 | 0.216 | **0.749** | -0.022 | 0.095 |
| 16 | 0.396 | 0.306 | **0.563** | 0.018 | 0.274 |
| 17 | 0.247 | 0.074 | -0.106 | 0.278 | **0.718** |
| 18 | 0.367 | 0.209 | -0.025 | 0.243 | **0.498** |
| 19 | -0.102 | 0.177 | 0.218 | 0.125 | **0.687** |
| 20 | 0.073 | 0.13 | 0.08 | **0.928** | 0.106 |
| 21 | 0.145 | 0.207 | 0.085 | **0.894** | 0.092 |
| 22 | 0.075 | 0.085 | 0.004 | **0.856** | 0.241 |
| 23 | 0.221 | **0.775** | 0.06 | 0.081 | 0.065 |
| 24 | 0.29 | **0.736** | 0.127 | 0.143 | -0.039 |
| 25 | 0.194 | **0.719** | -0.048 | 0.215 | 0.106 |
| 26 | 0.097 | **0.795** | 0.136 | 0.034 | 0.14 |
| 27 | 0.109 | **0.773** | 0.296 | 0.018 | 0.113 |
| 28 | 0.197 | **0.807** | 0.206 | 0.118 | -0.001 |
| 29 | 0.099 | **0.781** | 0.145 | 0.036 | 0.167 |

observed between COS score and medication score. Correlation between COS and GSH score was negative and close to conventional level of significance. There was no statistically significant correlation between skin, eyes, GHS Eye and COS score.

## Discussion

The original English version of Kings Sarcoidosis Questionnaire, with its five modules, had shown to be useful modular health status measure to be used in everyday clinical practice. Being a multimodular tool, depending on the afflicted organs, the physician can apply certain, or all modules in evaluating the state of the patient. The General Health Status module could be utilized in all forms of the disease, and, although being the most nonspecific, could be used in comparing different subpopulations of patients with sarcoidosis. During the creation of the Serbian version, we have appropriated all 29 items of the original version into Serbian language, and in doing so, we have allowed the Serbian version to be utilized in further potential cross-center studies. The internal reliability analyzed using Cronbach's α coefficient had

**Table 3. Correlation between the Serbian version of KSQ scores and other questionnaires.**

|  | SGRQ - SYMPTOMS | SGRQ - ACTIVITY | SGRQ - IMPACTS | SGRQ - TOTAL SCORE | Score DAL | Score MRC | FAS total | Score15D |
|---|---|---|---|---|---|---|---|---|
| **GHS score** | | | | | | | | |
| r | -0.523 | -0.623 | -0.588 | -0.663 | -0.668 | -0.567 | -0.739 | -0.699 |
| p | <0.001 | <0.001 | <0.001 | <0.001 | <0.001 | <0.001 | <0.001 | <0.001 |
| **Lung score** | | | | | | | | |
| r | -0.576 | -0.704 | -0.677 | -0.751 | -0.742 | -0.623 | -0.688 | -0.623 |
| p | <0.001 | <0.001 | <0.001 | <0.001 | <0.001 | <0.001 | <0.001 | <0.001 |
| **Med score** | | | | | ns | ns | | |
| r | -0.350 | -0.175 | -0.303 | -0.284 | | | -0.292 | -0.284 |
| p | <0.001 | 0.036 | <0.001 | 0.001 | | | <0.001 | 0.001 |
| **Skin score** | ns | ns | | ns | ns | ns | | |
| r | | | -0.157 | | | | -0.290 | -0.172 |
| p | | | 0.051 | | | | <0.001 | 0.032 |
| **Eyes score** | | | | | | | | |
| r | -0.382 | -0.400 | -0.405 | -0.445 | -0.380 | -0.410 | -0.538 | -0.574 |
| p | <0.001 | <0.001 | <0.001 | <0.001 | .002 | <0.001 | <0.001 | <0.001 |
| **GHS lung score** | | | | | | | | |
| r | -0.579 | -0.698 | -0.664 | -0.743 | -0.747 | -0.621 | -0.767 | -0.714 |
| p | <0.001 | <0.001 | <0.001 | <0.001 | <0.001 | <0.001 | <0.001 | <0.001 |
| **GHS skin score** | | | | | | | | |
| r | -0.512 | -0.594 | -0.580 | -0.644 | -0.646 | -0.535 | -0.743 | -0.690 |
| p | <0.001 | <0.001 | <0.001 | <0.001 | <0.001 | <0.001 | <0.001 | <0.001 |
| **GHS eyes score** | | | | | | | | |
| r | -0.512 | -0.595 | -0.558 | -0.633 | -0.561 | -0.554 | -0.736 | -0.740 |
| p | <0.001 | <0.001 | <0.001 | <0.001 | <0.001 | <0.001 | <0.001 | <0.001 |
| **GHS LS score** | | | | | | | | |
| r | -0.560 | -0.668 | -0.650 | -0.721 | -0.715 | -0.591 | -0.773 | -0.709 |
| p | <0.001 | <0.001 | <0.001 | <0.001 | <0.001 | <0.001 | <0.001 | <0.001 |
| **GHS LM score** | | | | | | | | |
| r | -0.579 | -0.646 | -0.656 | -0.713 | -0.746 | -0.576 | -0.744 | -0.689 |
| p | <0.001 | <0.001 | <0.001 | <0.001 | <0.001 | <0.001 | <0.001 | <0.001 |
| **GHS SM score** | | | | | | | | |
| r | -0.513 | -0.526 | -0.567 | -0.604 | -0.597 | -0.472 | -0.705 | -0.651 |
| p | <0.001 | <0.001 | <0.001 | <0.001 | <0.001 | <0.001 | <0.001 | <0.001 |
| **GHS LMS score** | | | | | | | | |
| r | -0.555 | -0.611 | -0.636 | -0.683 | -0.710 | -0.546 | -0.745 | -0.686 |
| p | <0.001 | <0.001 | <0.001 | <0.001 | <0.001 | <0.001 | <0.001 | <0.001 |

GHS- General Health Status, LS- Lung Skin, LM- Lung Medication, SM- Skin Medication, LSM- Lung Skin Medication, ns- non significant, r- Spearman correlation coefficient, SGRQ- Saint George Respiratory Questionnaire, DAL- Daily Activity List, MRC- Medical Research Council dyspnea scale, FAS- Fatigue Assessment Scale

shown similar results to the English version, as seen in the results section, which further proves the utility of Serbian version.

We have further compared the Serbian version of KSQ with other, standardized, questionnaires, and have shown strong correlation in all fields. SGRD and 15D are general quality of

**Table 4. Internal reliability for KSQ translated version.**

|  | KSQ modules | | | | |
|---|---|---|---|---|---|
|  | General health status | Lung | Skin | Eye | Medication |
| Number of items | 10 | 6 | 3 | 7 | 3 |
| Cronbach's α coefficient (Original KSQ) | 0.93 | 0.86 | 0.84 | 0.88 | 0.70 |
| Cronbach's α coefficient (Translated version of KSQ) | 0.90 | 0.91 | 0.92 | 0.91 | 0.67 |

KSQ- King's sarcoidosis questionnaire

**Table 5. Baseline characteristics of the study participants.**

| *Variables* | N (%) |
| --- | --- |
| Age (years), mean±sd (95% Confidence Interval) | 49.67±11.12 (47.83–51.56) |
| Gender n (%) | |
| Female | 107 (67.3) |
| Male | 52 (32.7) |
| Smoking n (%) | |
| Non smoker | 113 (71.1) |
| Ex smoker or smoker | 46 (28.9) |
| Stage of the lung disease n (%) | |
| 0 | 43 (27.2%) |
| 1 | 69 (43.0%) |
| 2 | 37 (23.4%) |
| 3 | 10 (6.3%) |
| Course of sarcoidosis n (%) | |
| Acute | 47 (26.9%) |
| Chronic | 112 (70.4%) |
| Extrapulmonary sarcoidosis n (%) | |
| Yes | 45 (28.3%) |
| No | 114 (71.7%) |
| Different organs involvement n (%) | |
| Skin | 8 (5.0%) |
| Eyes | 3 (1.9%) |
| Liver | 3 (1.9%) |
| Lymph node | 6 (3.8%) |
| Spleen | 4 (2.5%) |
| Heart | 2 (1.2%) |
| CNS | 19 (12%) |
| Medication n (%) | |
| No therapy | 71 (44.7%) |
| Prednisone | 73 (45.9%) |
| Methotrexat | 5 (3.1%) |
| Prednisone and Methotrexat | 10 (6.3%) |
| sACE U/L, mean±sd (95% Confidence Interval) | 51.32± 27.45 (46.61–56.03) |
| Lung function, mean±sd (95% Confidence Interval) | |
| FVC (liter) | 3.71±1.08 (3.49–3.84) |
| FVC (%) | 105.50±16.94 (102.89–108.17) |
| $FEV_1$ (liter) | 2.90±0.90 (2.73–3.02) |
| $FEV_1$ (%) | 98.44±18.93 (94.87–101.15) |
| $FEV_1$/FVC | 77.45±10.26 (75.93–79.30) |
| PEF (liter) | 7.41±2.54 (7.04–7.86) |
| PEF (%) | 102.69±21.62 (99.94–106.74) |
| DCo | 8.02±5.88 (6.99–9.10) |
| DCo (%) | 84.36±15.86 (81.51–87.01) |
| KCo | 1.47±0.21 (1.44–1.52) |

CNS- central nervous system, ACE- angiotensin converting enzyme, FVC- functional vital capacity, FEV1- forced expiratory volume in fist second, PEF- peak expiratory flow, DCo- diffusion lung capacity for carbon-monoxide, KCo- Carbon monoxide transfer coefficient

**Table 6. KSQ scores in patients with different clinical course of sarcoidosis.**

| | N | mean±sd | p |
|---|---|---|---|
| GHS score | | | |
| Acute | 46 | 66.9±21.3 | 0.213 |
| Chronic | 112 | 62.1±23.6 | |
| Lung score | | | |
| Acute | 46 | 75.8±19.8 | 0.013 |
| Chronic | 112 | 66.2±25.2 | |
| Medication score | | | |
| Acute | 46 | 73.8±17.6 | 0.124 |
| Chronic | 99 | 67.9±26.7 | |
| Skin score | | | |
| Acute | 45 | 91.0±18.4 | 0.012 |
| Chronic | 110 | 81.3±27.8 | |
| Eyes score | | | |
| Acute | 45 | 83.4±18.8 | <0.001 |
| Chronic | 110 | 68.4±26.7 | |
| GHS lung score | | | |
| Acute | 46 | 70.5±18.8 | 0.058 |
| Chronic | 112 | 63.7±23.0 | |
| GHS skin score | | | |
| Acute | 46 | 70.5±18.9 | 0.045 |
| Chronic | 112 | 63.7±23.0 | |
| GHS Eyes score | | | |
| Acute | 45 | 73.9±15.7 | 0.003 |
| Chronic | 110 | 64.4±22.1 | |
| GHS Lung Skin score | | | |
| Acute | 45 | 74.1±16.2 | 0.015 |
| Chronic | 110 | 66.4±21.4 | |
| GHS Lung Medication score | | | |
| Acute | 44 | 70.5±16.7 | 0.047 |
| Chronic | 99 | 63.8±21.7 | |
| GHS Skin Medication score | | | |
| Acute | 43 | 72.8±15.3 | 0.049 |
| Chronic | 98 | 66.6±20.0 | |
| GSH Lung skin medication score | | | |
| Acute | 43 | 73.7±14.8 | 0.016 |
| Chronic | 98 | 62.2±20.5 | |

GHS- General Health status

life questionnaires, already in use with both sarcoidosis patients and with other chronic patients, and two of the most common symptoms, dyspnea and fatigue, have their own separate questionnaires (MRS and FAS, respectively). The advantage of KSQ is that all of these results could be generated from one questionnaire, which leads to improved communication with the patients.

KSQ scores generated in this study have shown a statistically significant difference between patients with acute and chronic forms of the disease; the patients with the chronic form have shown significantly worse results as compared to those with acute form, and the difference is

**Table 7. Clinical outcome status and KSQ.**

| Sarcoidosis patients observed during five years (74 patients) | | | | | |
|---|---|---|---|---|---|
| *Resolved* | *Minimal Disease* | *Persistent sarcoidosis* | | | |
| **COS 1 Never treated** | **COS 3 Never treated** | *No current therapy* | | *Current therapy* | |
| COS 2 No therapy > One year 2pts | COS 4 No therapy > One year 9pts | COS 5 Never treated | COS 6 No therapy > One year 21pts | No worsening prior year | COS 9 Worsening in prior year 14pts |
| | | | | COS 7 Asymptomatic 5pts | |
| | | | | COS 8 Symptomatic 23pts | |

COS- clinical outcome status

present virtually all items. As is to be expected, patients with the chronic form have a higher chance to develop more severe forms of the pulmonary sarcoidosis, as well as extra pulmonary forms, both of which lead to significant decrease in quality of life, especially stage IV pulmonary sarcoidosis, neurosarcoidosis and cardiac sarcoidosis.

It should be noted there is another commonly used questionnaire in everyday practice- Sarcoidosis Health Questionnaire (SHQ), developed prior to KSQ [23]; this questionnaire has 29 items, and is to be utilized in its entirety. The original authors of KSQ have already compared the two questionnaires, and have found certain limitations of SHQ: unlike KSQ, SHQ cannot be tailored to individual phenotypes, SHQ has fewer items that evaluate fatigue, medication and extra pulmonary organs, with no validation for the usage in skin and eye disease. It also should be noted that different methodology has been used in validation of the questionnaires: KSQ with Rasch analysis and SHQ with clinical impact methodology, concurrent validity was higher in KSQ, and a stronger relationship was present between KSQ and SGRQ, as well as KSQ and MES dyspnea score. KSQ has also shown a higher test/retest value, the same was not present in SHQ [10]. During the creation of the Serbian version, we have taken into consideration the previously noted differences between KSQ and SHQ, and have not compared Serbian versions of these tests.

KSQ has been validated in both German and Dutch languages [24, 25]. The authors from the Netherlands have shown good internal consistency for all KSQ, and intraclass correlation coefficients and Bland-Altman plots showed good repeatability of the KSQ, therefore the test was successfully validated for Dutch language [24]. The original English version had 29 models, and in order to fulfil the requirements of the Rasch model, but the German version had to reduce the total number of models to 24; however, the original 29 version can still be used to compare data to international version [25].

There are certain limitations during the creation of the Serbian version. As the original authors have noted, during the creation of quality-of-life questionnaires, it is possible that certain characteristics have not been taken into consideration, but can be an important segment in patient's life. The quantity of items is also of importance; it is difficult to create a questionnaire of the least number of items, as it could be used in everyday practice, which still maintains its validity. Another point of interest is the interpretation of side effects of therapy. It is not always possible to differentiate the impairment of quality of life caused by the side effects of treatment and the disease, as noted by the authors [10]. One of the important parts when validating any questionnaire is measuring the minimal clinical important difference (MCID) which is defined by the smallest amount of change that is relevant to patients and would

**Table 8. Correlation between lung function parameters, clinical outcome of sarcoidosis and KSQ.**

| KSQ | Lung function parameters | | | | Clinical outcome of sarcoidosis |
|---|---|---|---|---|---|
| | FVC (%) | $FEV_1$ (%) | PEF (%) | DLco (%) | |
| GHS score | Ns | ns | | | ns |
| r | | | 0.252 | 0.289 | |
| p | | | 0.001 | <0.001 | |
| Lung score | Ns | | | | |
| r | | 0.168 | 0.290 | 0.343 | -0.236 |
| p | | 0.034 | <0.001 | <0.001 | 0.042 |
| Medication score | ns | ns | ns | ns | |
| R | | | | | -0.369 |
| p | | | | | 0.002 |
| Skin score | ns | ns | ns | ns | ns |
| r | | | | | |
| p | | | | | |
| Eye score | Ns | ns | | | ns |
| r | | | 0.177 | 0.230 | |
| p | | | 0.027 | 0.004 | |
| GHS Lung score | Ns | ns | | | |
| r | | | 0.285 | 0.332 | -0.256 |
| p | | | <0.001 | <0.001 | 0.027 |
| GHS Skin score | Ns | ns | | | |
| r | | | 0.223 | 0.305 | -0.235 |
| p | | | 0.005 | <0.001 | 0.037 |
| GHS Eye score | Ns | ns | | | ns |
| r | | | 0.259 | 0.315 | |
| p | | | 0.001 | <0.001 | |
| GHS Lung Skin score | Ns | ns | | | |
| r | | | 0.259 | 0.345 | -0.245 |
| p | | | 0.001 | <0.001 | 0.037 |
| GHS Lung Medication score | Ns | ns | | | |
| r | | | 0.251 | 0.380 | -0.286 |
| p | | | 0.002 | <0.001 | 0.020 |
| GHS Skin Medication score | Ns | ns | | | |
| r | | | 0.185 | 0.343 | -0.289 |
| p | | | 0.027 | <0.001 | 0.020 |
| GHS Lung Skin Medication score | Ns | ns | | | |
| r | | | 0.227 | 0.387 | -0.283 |
| p | | | 0.007 | <0.001 | 0.002 |

KSQ- King's sarcoidosis questionnaire, GHS- General health status, ns- non significant, r- Pearson correlation coefficient, FVC- functional vital capacity, FEV1- forced expiratory volume in fist second, PEF- peak expiratory flow, DLCO- diffusion lung capacity for carbon-monoxide

indicate the success or detriment of an intervention or warrant changes in the care plan [25]. A study was performed where MCID was calculated for two major domains of the KSQ (the general health and pulmonary-specific). The study had taken into analysis several question-naires (KSQ, SGRQ, FAS and the Short Form-36) with physiological tests (FVC and six-min-ute-walk tests). They identified very weak correlations between KSQ scores and physiological (FVC and six-minute-walk tests) measures of the severity of pulmonary sarcoidosis. We have shown a strong correlation between physiological tests and KSQ score, however we have not performed a follow-up, and therefore we could not give our conclusion. A precise MCID has yet to be discovered for KSQ [26, 27].

## Conclusion

As the authors of the original study have shown, the questionnaire is a quick and valid tool to identify health status issues important to patients, and can be used to help formulate shared care plans between the patient and physician. The Serbian version of KSQ has shown an important step towards analyzing the quality of life in patients with sarcoidosis, and allows the comparison between centers internationally. Being translated, this version can be used by both physicians and patients that do not speak English, which will lead to the increase of data generation, and will lead to better understanding of quality of life in this disease. It should be noted that the questionnaire should not be used isolated, and should be improved with clinical presentation, radiological and functional tests and other questionnaires. As the original version undergoes reevaluation and improvement, so will the Serbian version.

## Supporting information

**S1 File.**
(SAV)

**S2 File.**
(DOCX)

**S3 File.**
(DOCX)

## Author Contributions

**Conceptualization:** Mihailo Stjepanovic, Violeta Mihailovic-Vucinic.

**Data curation:** Aleksa Golubovic.

**Formal analysis:** Branislav S. Gvozdenovic, Slobodan Belic.

**Funding acquisition:** Mihailo Stjepanovic.

**Investigation:** Mihailo Stjepanovic, Branislav S. Gvozdenovic, Slobodan Belic, Nikola Maric.

**Methodology:** Branislav S. Gvozdenovic, Nikola Maric.

**Resources:** Mihailo Stjepanovic, Jelena Milin-Lazovic, Aleksa Golubovic.

**Software:** Jelena Milin-Lazovic, Natasa Djurdjevic.

**Supervision:** Violeta Mihailovic-Vucinic, Natasa Djurdjevic.

**Validation:** Violeta Mihailovic-Vucinic, Jelena Milin-Lazovic.

**Visualization:** Natasa Djurdjevic.

**Writing – original draft:** Mihailo Stjepanovic.

**Writing – review & editing:** Violeta Mihailovic-Vucinic.

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
