## [Decision Letter · Decision Letter 0]

29 Aug 2022

PONE-D-22-20583King’s Sarcoidosis Questionnaire (KSQ) – Validation study in non-English speaking population of sarcoidosis patientsPLOS ONE

Dear Dr. Stjepanovic,

Thank you for submitting your manuscript to PLOS ONE. After careful consideration, we feel that it has merit but does not fully meet PLOS ONE’s publication criteria as it currently stands. Therefore, we invite you to submit a revised version of the manuscript that addresses the points raised during the review process.

We look forward to receiving your revised manuscript.

Kind regards,

Supat Chupradit, Ph.D., M.Ed., B.Sc.(OT), B.P.A., B.Ed., B.A.

Academic Editor

PLOS ONE

Journal Requirements:

4. Please amend your authorship list in your manuscript file to include author Aleksa Golubovic.

Additional Editor Comments:

Dear Author,

Please see reviewers suggest to you. You should improve by the comments and resubmit by deadline to our consider.

Regards

Academic Editor

Reviewers' comments:

Reviewer's Responses to Questions

**Comments to the Author**

1. Is the manuscript technically sound, and do the data support the conclusions?

Reviewer #1: Yes

Reviewer #2: No

Reviewer #3: Partly

Reviewer #4: Yes

Reviewer #5: Yes

Reviewer #6: Yes

Reviewer #7: Yes

Reviewer #8: Yes

Reviewer #9: Yes

Reviewer #10: Partly

2. Has the statistical analysis been performed appropriately and rigorously? 

Reviewer #1: Yes

Reviewer #2: No

Reviewer #3: No

Reviewer #4: Yes

Reviewer #5: Yes

Reviewer #6: Yes

Reviewer #7: Yes

Reviewer #8: Yes

Reviewer #9: Yes

Reviewer #10: No

3. Have the authors made all data underlying the findings in their manuscript fully available?

Reviewer #1: Yes

Reviewer #2: Yes

Reviewer #3: Yes

Reviewer #4: Yes

Reviewer #5: Yes

Reviewer #6: Yes

Reviewer #7: Yes

Reviewer #8: Yes

Reviewer #9: Yes

Reviewer #10: Yes

4. Is the manuscript presented in an intelligible fashion and written in standard English?

Reviewer #1: Yes

Reviewer #2: No

Reviewer #3: Yes

Reviewer #4: Yes

Reviewer #5: No

Reviewer #6: Yes

Reviewer #7: Yes

Reviewer #8: Yes

Reviewer #9: No

Reviewer #10: No

5. Review Comments to the Author

Reviewer #1: King’s Sarcoidosis Questionnaire (KSQ) – Validation study in non-English speaking population of sarcoidosis patients

. This submission is based on substantial research efforts, so it may be suitable for publication in Journal after revising. This article is strongly recommended for publication after incorporating certain changes. This article needs thorough proofreading. Overall quality of Language is good. Just minor grammatical mistakes are found. All tables and figures are relevant. Research Methodology has been well defined. All data are aligned to the findings of the research. This article is good attempt in the field research and will be beneficial for future researchers.

Reviewer #2: This manuscript will greatly benefit from proofreading and language editing.

1. Abstract: The Introduction section of the abstract is quite long. Even longer than the methods. Streamline your thought to introduce your topic and the gap of the literature. Be direct to only identify the objectives and any other specific objectives. In the methods, identify whether the instrument was self-answered, and what statistical analysis was performed.

2. The introduction lacks focus on the King’s Sarcoidosis Questionnaire. There was even a mention of another questionnaire that was irrelevant to the study. The authors will need to organize their thought process more and determine the need for using this questionnaire, its constructs, evidence of its psychometric properties, and the need to contextualize.

3. I am confused whether this is a psychometric study or a cross-sectional study. The aim does not match the methodology identified.

4. Include the specific ethical approval number and IRB information.

5. Restructure the Methods section. It is quite confusing how several similar information can be in separate subheadings.

6. What is the rationale behind the other questionnaires?

7. What was the reference used to inform the process translation of the questionnaire? Shouldn't this be the first phase of the study?

8. How is internal reliability in the results section different from internal consistency?

9. What was the basis for interpreting the patients' scores to be low on the KSQ?

10. Overall the results section was confusing. Once the authors explicitly describe their objective/s, this section may be better constructed following those for better clarity and coherence.

11. Limit your discussion to the study's findings grounded on your objectives.

12. What key messages on the KSQ are your pertaining to?

13. To claim the suitability in non-English speaking population is an over-inflation of your findings, which was contextualized to your setting. You do not have any proof that apart from the ethnicity/culture of the patients recruited in study, the same psychometric properties will be the same in others.

Reviewer #3: 1) The introduction part : the authors should identify the significant problem or advantage of Sarcoidosis symptom diagnostic in this article target.

2) According to examine the construct validity by using Spearman's correlation coefficient thus the authors should describe how to calculate the sampling size or describe the sampling technique. In addition, the authors should identify the criteria for selected the participant and reject the participant in this research.

3) Due to this research conduct with non English speaking the authors should examine the translate validation in any scale that is used in this research. Moreover, should examine the reliability by using test retest between English scale and translated scale.

4) According to this research construct validity by using Spearman's correlation coefficient that questionable seem like concurrent validity technique

Reviewer #4: The review article topic: King’s Sarcoidosis Questionnaire (KSQ) – Validation study in non-English speaking population of sarcoidosis patients. This submission is based on substantial research efforts, so it may be suitable for publication in Journal after revising. This article is strongly recommended for publication after incorporating certain changes. This article needs thorough proofreading. The overall quality of the Language is good. Just minor grammatical mistakes are found. All tables and figures are relevant. The research Methodology has been well defined. All data are aligned with the findings of the research. This article is a good attempt at field research and will be beneficial for future researchers.

Reviewer #5: Although the translation of the test into other societies with linguistic and cultural differences is important and appropriate for research to be published in international journals, But the composition of this article is still far from the standard of this journal. Authors need to revamp most of the content to be resubmitted.

Reviewer #6: Dear Authors,

This paper investigated the on PLOS ONE manuscript "King’s Sarcoidosis Questionnaire (KSQ) – Validation study in non-English speaking population of sarcoidosis patients." I think this paper needs minor revision to improve the rationale and gap of the study. Specific comments for each section are below:

Introduction

Page 4, I am not clear about why physicians and patients did not agree in the symptoms of sarcoidosis. Please clarify this point or add the issue related to symptoms between physicians and patients.

Page 4, Give relevant reasons why your team choose “the King’s Sarcoidosis Questionnaire (KSQ)” rather than the Sarcoidosis Health Questionnaire. You mention about 80% of them were African Americans in the study. Why will be affected your study?

Please showed outstanding points are the reasons KSO from England that would be possible to synthesize the aim of your study.

Methods

Page 8, The comprehensive survey, or questionnaire survey is important that you need to clarify how to translate the needed evidence or method to support it (Translation process), one mother tongue expert should be in the field of health professional.

Page 10, After obtaining the final Serbian version, you provided it to a pilot study (Five patients and five physicians). However, the next process will be the inclusion and exclusion criteria that I cannot see. You needed to show this relevant point on the method as well. For example, you need to set up the age rank of your patients, etc. It will help you to show the outcome of results in the demographic table (Table 3) in the result section.

Discussion

Page 14, your opinion further studies are necessary to distinguish the components of fatigue (mental and physical) in correlation with KSQ. You mentioned these points, please clarify why the future KSQ Serbian version should be distinguished the physical and mental fatigues in this questionnaire.

Page 14, Please rewrite “The German version required the reduction of models, from original 29 to 24, to fulfill the requirements of the Rasch model; however, the original 29 version can still be used to compare data to the international version [36].”

The readers may be confused about “29 to 24”, needs to be clarified what is it (items). And “the original 29 version”, please rewrite to be clear.

Reviewer #7: Overall, a well-written article that could use some polishing in terms of packaging and flow. The authors may also want to add research implications.

Methods:

Sampling methodology needs elaboration.

The research method is explained in detail and is easy to understand.

Testing the validity and reliability of the instrument can be used.

Results:

The results of the study have been well explained by the author and are supported by valid primary data and appropriate conclusions.

Reviewer #8: Abstract

- Results: summary of statistic analysis should be added.

Objectives: The author should use the word consistency (aim or objective).

Methods: Re-arrange this section.

Patient: If patients incompleted the questionnaire, were they withdrawn from the study?

Add number of study approved.

There is no need to write doctors and patients' comments.

Results: Re-arrange this section.

Discussion: Begin with a clear statement.

Be concise.

Reviewer #9: 1) Abstract

- please paraphrase this sentence - Despite the studies performed

with the aim to find out the reliable tools for assessment the disease severity or its

progression, up to date there is no reliable tool for assessing the impact of sarcoidosis on

patients’ health status and the impact of the therapy.

- Sarcoidosis health Questionnaire, created by Christopher Cox and coworkers (year?)

2) Introduction

- should highlight the need of a new assessment for measuring health status in sarcoidosis.

- what are the weakness of the current assessment.

3) Methods

- authors should rewrite the way they present the translation process espcially re doctors' comments to a more sound academic writing.

Reviewer #10: Comment to authors

This article provides the validation results of the King’s Sarcoidosis Questionnaire (KSQ) in the non-English speaking population of sarcoidosis patients. The manuscript has limitations on language usage as there were a lot of non-academic and awkward sentences. It would be good if the authors could revise the manuscript to make it more precise and constructive. In my opinion, this paper requires significant revision. I hope my comments and suggestions below are helpful.

• The authors should reconsider the title of this research to be specific to the sample of the current study (Serbian) instead of using non-English speaking.

• Abstract; no need to mention another questionnaire that was not used in the current study.

Introduction

• The Introduction is too short. The authors should provide more rationale for research gap.

• A comprehensive literature review should be provided to reveal the correlation between the quality of life and sarcoidosis patients’ life

• Please provide the rationale for choosing the King’s Sarcoidosis Questionnaire (KSQ) over the Sarcoidosis Health Questionnaire.

• Please add more information on previous cross-cultural studies that examined the psychometric properties of the KSQ.

Methods

• Please give detail on the inclusion and exclusion criteria used. It’s not clear in the methods.

• Please provide more information on sample recruitment and the rational of enrolling samples 18 years and older.

• Please revise the Methods section to be more structured and clearer, particularly on Questionnaire

• ‘Questionnaire’ should be on the sub-heading not ‘ questionnaire used in this study’

• Please provide psychometric properties, i.e. internal consistency of all questionnaires used in the current study

• In the back translation process, two bilinguals who expertise both in English and Serbian are needed, not only two mother tongue experts

• No need to mention the full name of the person who scored the KSQ on the manuscript

• In my opinion, the authors should provide only the Item-Objective Congruence (IOC) results instead of giving all doctors’ comments on the manuscript

Statistical analysis

• Please elaborate statistical analysis more precise and clearly

• Please provide how missing data was handled

• Description of results of normality of distribution should be evident in the section

Results

• There are many questionnaires used in the current study and also an abbreviation, therefore the authors should consider reporting results more precise and clearly by using table

Discussion

• The first paragraph of the Discussion should be moved to Introduction

• No need to repeat all the results of the discussion.

• The discussion needs to improve and add information on the strengths and limitations of the study.

• Please add the information about the clinical application of this study.

• As this current study examines convergent validity and internal consistency, it could not be concluded that the KSQ is a reliable and valid tool. The findings should be interpreted with caution regarding the limitation. Other psychometric properties, e.g. test-retest reliability, criterion validity, and construct validity of the KSQ should also be examined.

6. PLOS authors have the option to publish the peer review history of their article (what does this mean?). If published, this will include your full peer review and any attached files.

Reviewer #1: No

Reviewer #2: No

Reviewer #3: No

Reviewer #4: No

Reviewer #5: **Yes: **Kittisak JERMSITTIPARSERT

Reviewer #6: No

Reviewer #7: No

Reviewer #8: **Yes: **Natwipa Wanicharoen, PhD.

Reviewer #9: No

Reviewer #10: No

---

## [Author Response · Author response to Decision Letter 0]

1 Nov 2022

To whom it may concern,

We have taken into consideration the points raised by the academic editor and reviewers, and have corrected accordingly.

• The majority of the article has been rewritten, as to improve the quality of language.

• The title of the article has been changed, as to be more specific (from “non-English” to “Serbian”).

• We have removed the other questionnaire that was not used in the study from the Abstract.

• We have expanded the introduction as to emphasize the need for a quality-of-life questionnaire.

• The ethical aspect, as well as the inclusion and exclusion criteria have been expanded upon in material and methods section.

• Other questionnaires that have been used for comparison have been explained as well as rationale behind using them in the material and methods section.

• We have removed the results from the discussion section.

• The limitations of the study have been noted in the discussion section.

• Clinical application of the questionnaire has been explained in the conclusion.

• The versions of KSQ in other languages were discussed, and the rationale for using the KSQ over Sarcoidosis Health Questionnaire was explained.

We hope that our changes to the article will suffice, and that our altered article will meet the criteria for publication. 

The autors

---

## [Decision Letter · Decision Letter 1]

22 Nov 2022

PONE-D-22-20583R1King’s Sarcoidosis Questionnaire (KSQ) – Validation study in Serbian speaking population of sarcoidosis patientsPLOS ONE

Dear Dr. Stjepanovic,

Thank you for submitting your manuscript to PLOS ONE. After careful consideration, we feel that it has merit but does not fully meet PLOS ONE’s publication criteria as it currently stands. Therefore, we invite you to submit a revised version of the manuscript that addresses the points raised during the review process.

We look forward to receiving your revised manuscript.

Kind regards,

Supat Chupradit, Ph.D., M.Ed., B.Sc.(OT), B.P.A., B.Ed., B.A.

Academic Editor

PLOS ONE

Journal Requirements:

Reviewers' comments:

Reviewer's Responses to Questions

**Comments to the Author**

1. If the authors have adequately addressed your comments raised in a previous round of review and you feel that this manuscript is now acceptable for publication, you may indicate that here to bypass the “Comments to the Author” section, enter your conflict of interest statement in the “Confidential to Editor” section, and submit your "Accept" recommendation.

Reviewer #1: All comments have been addressed

Reviewer #2: (No Response)

Reviewer #3: All comments have been addressed

Reviewer #4: All comments have been addressed

Reviewer #6: All comments have been addressed

Reviewer #7: All comments have been addressed

Reviewer #8: All comments have been addressed

2. Is the manuscript technically sound, and do the data support the conclusions?

Reviewer #1: Yes

Reviewer #2: Partly

Reviewer #3: Yes

Reviewer #4: Yes

Reviewer #6: Yes

Reviewer #7: Yes

Reviewer #8: Yes

3. Has the statistical analysis been performed appropriately and rigorously? 

Reviewer #1: Yes

Reviewer #2: No

Reviewer #3: No

Reviewer #4: Yes

Reviewer #6: Yes

Reviewer #7: Yes

Reviewer #8: Yes

4. Have the authors made all data underlying the findings in their manuscript fully available?

Reviewer #1: Yes

Reviewer #2: Yes

Reviewer #3: Yes

Reviewer #4: Yes

Reviewer #6: Yes

Reviewer #7: Yes

Reviewer #8: Yes

5. Is the manuscript presented in an intelligible fashion and written in standard English?

Reviewer #1: Yes

Reviewer #2: No

Reviewer #3: Yes

Reviewer #4: Yes

Reviewer #6: Yes

Reviewer #7: Yes

Reviewer #8: Yes

6. Review Comments to the Author

Reviewer #1: King’s Sarcoidosis Questionnaire (KSQ) – Validation study in Serbian speaking population of sarcoidosis patients. Overall clear and follow by reviewer comments. Accept

Reviewer #2: It is quite difficult to re-review the manuscript without the authors' point-by-point response to my previous comment. It is for this mater that I retain my previous comments on the manuscript.

Reviewer #3: 1-The authors aim to validate the construct validity of research tools. The construct validity that might be found by using Exploratory factor analysis and comfirmatory factor analysis. In another way we found by using expert judgement . The Pearson product moment correlation is not the method for construct validation. This method is used only for concurrent validation or predictive validation in research tools. The authors should consider the method for found the psychometic propertrict of reserch tools.

2-We found the negative correration and lower correlation. Therefore, the authors might describe indept why and how was ??

3-the method of sampling size for using Pearson correlation should calculate the power of test, error, effectsize, and level of statistically significant levels.

Reviewer #4: Thank you for considering reviewer comments and suggestions. I am satisfied with the responses.

All the best for your article.

Reviewer #6: Dear Authors,

This paper investigated the PLOS ONE manuscript "King’s Sarcoidosis Questionnaire (KSQ) – Validation study in Serbian speaking population of sarcoidosis patients."

The manuscript revision has been improved; all the queries have been answered appropriately. The authors had already responded to the main concerns and relevant issues. I found this work to be persuasive and consistent with scholarly intent and intellectual passion to contribute to supporting sarcoidosis patients in Serbia.

One recommendation for further study would be to utilize the treatment (before-after), which can strengthen your work and benefit sarcoidosis patients in Serbia.

I recommended this work for publication or no revision.

Thank you very much

Reviewer

Reviewer #7: It is encouraged to publish this article after making certain adjustments. The article need a thorough proofread. Frequently, the language is of high quality. Minor grammatical errors are the only ones that have been found. Each graph and table is relevant. Research technique is clearly defined. The findings of the study agree with all of the information. The excellent research effort put out in this essay will be advantageous to upcoming researchers.

Reviewer #8: all comments have been addressed. The authors have adequately addressed comments raised in a previous round of review and you feel that this manuscript is now acceptable for publication

7. PLOS authors have the option to publish the peer review history of their article (what does this mean?). If published, this will include your full peer review and any attached files.

Reviewer #1: No

Reviewer #2: No

Reviewer #3: No

Reviewer #4: No

Reviewer #6: No

Reviewer #7: No

Reviewer #8: No

---

## [Author Response · Author response to Decision Letter 1]

5 Jan 2023

As requested, all of the changes were implemented in the paper, with specifics noted in the Response to Reviewers file

---

## [Decision Letter · Decision Letter 2]

6 Mar 2023

PONE-D-22-20583R2

King’s Sarcoidosis Questionnaire (KSQ) – Validation study in Serbian speaking population of sarcoidosis patients

PLOS ONE

Dear Dr. Stjepanovic,

Thank you for submitting your manuscript to PLOS ONE. After careful consideration, we feel that it has merit but does not fully meet PLOS ONE’s publication criteria as it currently stands. Therefore, we invite you to submit a revised version of the manuscript that addresses the points raised during the review process.

I am sorry, but I stepped into this review process, as the former Academic Reviewer is no longer available. Therefore I do not have full knowledge on the history of this manuscript. To me it seems that most of the concerns are sufficiently answered; however, reviewer 2 still has some open questions to be clarified. Thus, I would like to ask you to amend your manuscript according to the reviewer's suggestions wherever possible and necessary. Please send us a detailed point-by-point answer to the reviewers comments.

We look forward to receiving your revised manuscript.

Kind regards,

Supat Chupradit, Ph.D., M.Ed., B.Sc., B.P.A., B.Ed., B.A.

Academic Editor

PLOS ONE

Journal Requirements:

Reviewers' comments:

Reviewer's Responses to Questions

**Comments to the Author**

1. If the authors have adequately addressed your comments raised in a previous round of review and you feel that this manuscript is now acceptable for publication, you may indicate that here to bypass the “Comments to the Author” section, enter your conflict of interest statement in the “Confidential to Editor” section, and submit your "Accept" recommendation.

Reviewer #2: (No Response)

Reviewer #3: All comments have been addressed

2. Is the manuscript technically sound, and do the data support the conclusions?

Reviewer #2: Partly

Reviewer #3: Yes

3. Has the statistical analysis been performed appropriately and rigorously? 

Reviewer #2: Yes

Reviewer #3: Yes

4. Have the authors made all data underlying the findings in their manuscript fully available?

Reviewer #2: Yes

Reviewer #3: Yes

5. Is the manuscript presented in an intelligible fashion and written in standard English?

Reviewer #2: Yes

Reviewer #3: Yes

6. Review Comments to the Author

Reviewer #2: 1. The rationale behind the need to test KSQ in a Serbian population should be placed in the Introduction and not in the methods section.

2. Consider placing the estimated COS groups in a table.

3. Why was back translation not performed prior to cognitive interviewing?

4. How did the authors resolved the rating of (3) made by the last participant?

5. How robust was the sample size? Conducting factor analysis usually recruits 20--300 participants or 10 participants x number of items.

7. Were the five components used in the EFA correspond to the five modules of the KSQ?

8. Why was an overall internal consistency cronbach's alpha not computed?

9.

Reviewer #3: well revise version

the authors should design the data table that should consist the only significant data.

7. PLOS authors have the option to publish the peer review history of their article (what does this mean?). If published, this will include your full peer review and any attached files.

Reviewer #2: No

Reviewer #3: No

---

## [Author Response · Author response to Decision Letter 2]

8 Apr 2023

We hope that we have successfully responded to your suggestions, and that our paper is now up to PLOS ONE standards. All of the changes are noted in the manuscript with track changes, and in a separate file, as requested.

---

## [Decision Letter · Decision Letter 3]

24 Apr 2023

King’s Sarcoidosis Questionnaire (KSQ) – Validation study in Serbian speaking population of sarcoidosis patients

PONE-D-22-20583R3

Dear Dr. Stjepanovic,

We’re pleased to inform you that your manuscript has been judged scientifically suitable for publication and will be formally accepted for publication once it meets all outstanding technical requirements.

Kind regards,

Gernot Zissel, Ph.D.

Academic Editor

PLOS ONE

Additional Editor Comments (optional):

Reviewers' comments:

Reviewer's Responses to Questions

**Comments to the Author**

1. If the authors have adequately addressed your comments raised in a previous round of review and you feel that this manuscript is now acceptable for publication, you may indicate that here to bypass the “Comments to the Author” section, enter your conflict of interest statement in the “Confidential to Editor” section, and submit your "Accept" recommendation.

Reviewer #2: All comments have been addressed

2. Is the manuscript technically sound, and do the data support the conclusions?

Reviewer #2: Yes

3. Has the statistical analysis been performed appropriately and rigorously? 

Reviewer #2: Yes

4. Have the authors made all data underlying the findings in their manuscript fully available?

Reviewer #2: Yes

5. Is the manuscript presented in an intelligible fashion and written in standard English?

Reviewer #2: Yes

6. Review Comments to the Author

Reviewer #2: (No Response)

7. PLOS authors have the option to publish the peer review history of their article (what does this mean?). If published, this will include your full peer review and any attached files.

Reviewer #2: No

---

## [Editor Report · Acceptance letter]

25 Aug 2023

PONE-D-22-20583R3 

King’s Sarcoidosis Questionnaire (KSQ) – Validation study in Serbian speaking population of sarcoidosis patients 

Dear Dr. Stjepanovic:

I'm pleased to inform you that your manuscript has been deemed suitable for publication in PLOS ONE. Congratulations! Your manuscript is now with our production department. 

Kind regards, 

on behalf of

Prof. Dr. Gernot Zissel 

Academic Editor

PLOS ONE